# Spatial Pattern Evolution and Influencing Factors of Tourism Flow in the Chengdu–Chongqing Economic Circle in China

**Xuejun Chen** [1] , **Yang Huang** [2,*] **and Yuesheng Chen** [3]

1    College of Tourism and Media, Chongqing Jiaotong University, Chongqing 400074, China; 990020607064@cqjtu.edu.cn
2    Chinese Village Culture Research Center, Central South University, Changsha 410083, China
3    School of Artificial Intelligence, Beijing University of Posts and Telecommunications, Beijing 100876, China; visen@bupt.edu.cn
*    Correspondence: ocean520@csu.edu.cn; Tel.: +86-19307492616

**Abstract:** Based on Ctrip's 'tourism digital footprint', the spatial pattern of tourism flows in the Chengdu–Chongqing Economic Circle from 2018 to 2021 is explored, social network analysis and spatial visualisation of tourism information data are conducted, and factors affecting the network structure of tourism flows are analysed using linear weighted regression methods. The results show that tourism flows in the Chengdu–Chongqing Economic Circle show a significant 'dual core' polarisation effect. At the end of 2019, as a turning point, the density value of the tourism flow network shows an irregular inverted 'U' distribution. Kuanzhai Alley, Hong Ya Dong and Chunxi Road have irreplaceable competitive advantages in the tourism flow network. The density of highways, the number of star-rated hotels and the regional GDP per capita are positively correlated with the effective size of the structural hole of the administrative unit. Finally, based on the research results, countermeasures are proposed to optimise the tourism development of the Chengdu–Chongqing Economic Circle.

**Keywords:** tourism flow; social network analysis; spatial pattern; influencing factors





## 1. Introduction

China has five major city clusters: the Chengdu–Chongqing City Cluster, the Yangtze River Delta City Cluster, the Guangdong–Hong Kong–Macao Greater Bay Area, the Beijing–Tianjin–Hebei City Cluster and the Yangtze River Midstream City Cluster. Among them, Chengdu–Chongqing City Cluster is the only national city cluster in the western inland of China. The Chengdu–Chongqing Economic Circle plays a key role in China's regional development; the importance of its tourism industry is also growing. The public is gradually getting used to obtaining online tourism information and recording travel experiences. The resulting 'digital footprint of tourism' provides a huge database for scholars to study tourism flows. The destination network composed of tourism flow is of great value for tourism research in urban agglomerations. Studying the distribution, flow law and influencing factors of tourism flows in the Chengdu–Chongqing Economic Circle has important practical guidance significance for promoting the coordinated development of the tourism industry in the western inland region of China.

## 2. Related Works

Foreign research on tourism flows began in the 1960s [1], with studies on the conceptual system of tourism flows [2], the spatial and temporal evolution of tourism flows [3], the drivers of tourism flows [4,5] and the flow effects of tourism flow [6]. For example, Salas-Olmedo et al. analysed the 'digital tourism footprint' of urban tourists through big data using three data sources to reflect the spatial distribution characteristics of tourism activities of sightseeing tourists, consumer tourists and accommodation tourists in Madrid [7].

Therefore, the 'tourism digital footprint' has been widely used in academia to study the spatial distribution of regional tourism flows. Research on domestic tourism flows started in the 1980s [1], covering various topics, such as the conceptual system of tourism flows [8], the spatial and temporal evolution of tourism flows [9–11], the flow effects of tourism flows [12] and the network structure of tourism flows [13–17]. Researchers usually select continuous or spaced study periods using a 'tourism digital footprint' when studying the spatial pattern evolution of tourism flows. The spatial scale of the study is mainly specific tourism areas, cities, provinces, countries and continents. For example, Bai Gang et al. used Weibo tourism data to analyse the spatial and temporal changes in domestic tourism flows in Lijiang River Scenic Area and Yangshuo Scenic Area in Guilin City from 2016 to 2019 and quantitatively compared the spatial and temporal evolution differences of tourism flows between the two places [18]. When studying the influencing factors of the spatial pattern evolution of tourism flows, scholars mostly conduct quantitative research by calculating the index correlation [19], constructing an index system [20], building a regression model [21] and clustering analysis by groups [22]. For example, Wang Yongming et al. selected eight indicators from six dimensions—economic development, traffic conditions, tourism resources, tourism facilities, government support and cultural factors—to build an index system of the influencing factors of the spatial network of tourism flows [20]. These studies also provide the article with a rich methodology for exploring the factors influencing the development characteristics of the spatial pattern of tourism flows in the Chengdu–Chongqing Economic Circle. Domestic and international tourism flow-related research results are rich, research fields are wide, and research methods are constantly evolving, providing a solid theoretical foundation for this study. However, scholars have conducted few relevant studies on the specific region of the Chengdu–Chongqing Economic Circle.

This article takes the only national city cluster in the western inland of China as the research subject and studies the development of its tourism flow network spatial structure in an ephemeral research method, which has a certain academic innovation value. This article is closely related to the 'Outline of the Construction of Chengdu–Chongqing Economic Circle' and the actual situation of tourism development in the Chengdu–Chongqing Economic Circle, proposes countermeasures to optimise tourism development in the Chengdu–Chongqing Economic Circle and lays a solid theoretical foundation for tourism development in the Chengdu–Chongqing urban agglomeration.

## 3. Study Area, Research Methods and Data Processing

### 3.1. Study Area

The planning area of the Chengdu–Chongqing Economic Circle includes the central city of Chongqing and 27 districts (counties), including Wanzhou, Fuling, Qijiang, Dazu, Qianjiang, Changshou, Jiangjin, Hechuan, Yongchuan, Nanchang, Bishan, Tongliang, Tongnan, Rongchang, Liangping, Fengdu, Dianjiang, Zhongxian and parts of Kaizhou and Yunyang, and 15 cities in Sichuan Province, namely, Chengdu, Zigong, Luzhou, Deyang, Mianyang (except Pingwu County and Beichuan County), Suining, Neijiang, Leshan, Nanchong, Meishan, Yibin, Guang'an, Dazhou (except Wanyuan City), Ya'an (except Tianquan County, Boxing County) and Ziyang [23]. The yellow patches in Figure 1 correspond to the Chongqing part of the Chengdu–Chongqing Economic Circle, the red patches correspond to the Sichuan part, the corresponding province names are marked in blue, and the cities and regions within the Chengdu–Chongqing Economic Circle are in black. The 44 cities or counties in the planning area are referred to below as administrative units.

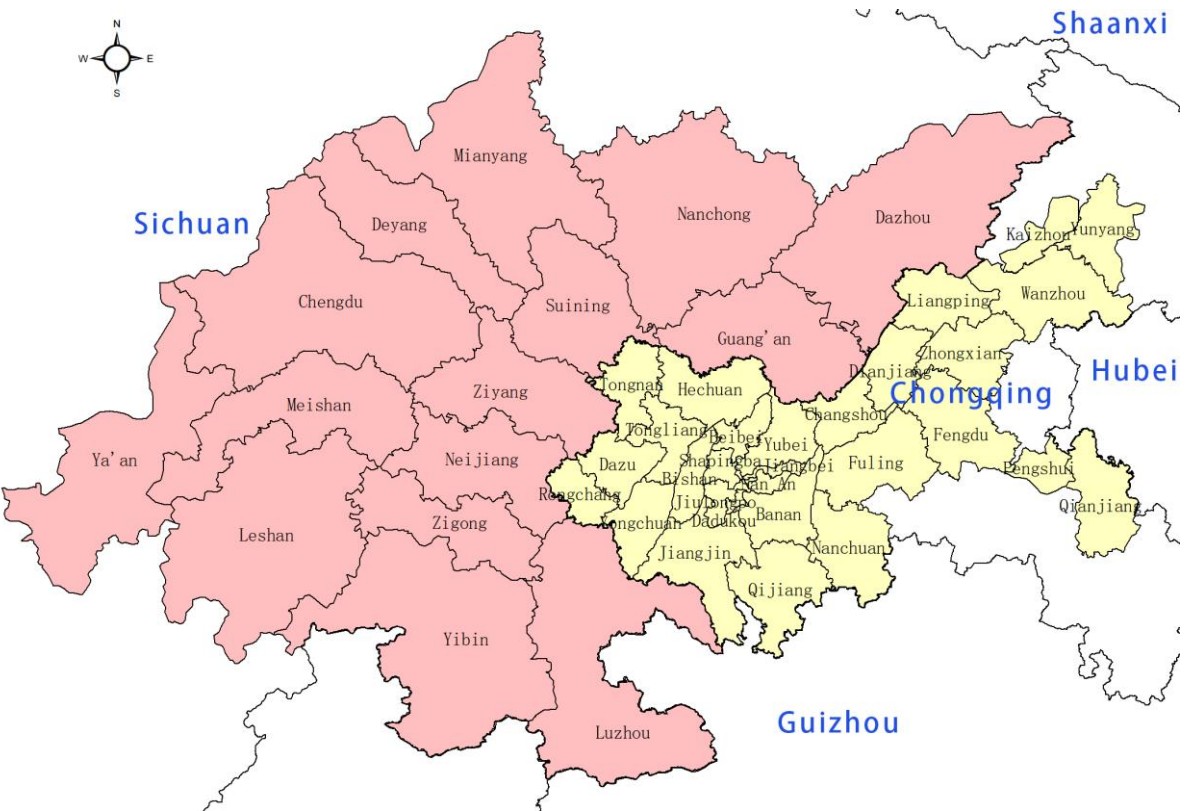

**Figure 1.** Map of the Chengdu–Chongqing Economic Circle.

*3.2. Research Methods*

3.2.1. Social Network Analysis

Social network analysis is a sociological method, which was proposed by Alfred Radcliffe-Brown, a famous British anthropologist in the mid-20th century, in his study of structural features. [24] In 1992, Ronald Burt proposed the theory of 'structural holes' in his book *Structural Holes: The Social Structure of Competition*, which refers to the gap created by the lack of connection between two individuals in a network who have complementary sources of information [25]. The social network analysis method has been applied to study tourism flows and can accurately reveal the structural characteristics of tourism networks. It is a common method used by scholars to study tourism flows [26–35], including the Spanish scholar Gonzalez-Diaz, who has used it to study the dynamics of tourist flows concerning the configuration of accommodation networks in Spanish regions [26]. Based on a review of research studies in the literature [36–38], Gephi0.9.5 and ArcGIS10.5 software were used to draw the tourism flow network of the Chengdu–Chongqing Economic Circle for 2018–2021. UCINET6.0 was used to analyse the structural characteristics and spatial pattern development trend of the tourism network in the Chengdu–Chongqing urban agglomeration from the perspective of network density, centrality and structural holes. The indicators covered in this section and their calculation formulas are as follows.

Professor Liu Jun, an early Chinese scholar engaged in social network research, made a systematic and detailed explanation of the social network analysis method, network density, centrality and structural holes in the overall network analysis handout and gave the corresponding applicable scenarios and calculation method [37]. Chinese scholars Wang Shuhua et al. also explained the indicators in Table 1 and showed calculation methods for the structural characteristics of tourism flow networks in Henan Province [38].

**Table 1.** Indicators of the structure of the tourism flow network and their meaning [36–38].

| Indicators | Meaning | Calculation Formula or Realization Path |
|---|---|---|
| Network Density | The ratio of the actual number of relationships present in the trip flow network to the theoretical maximum number of relationships present. | This is achieved through the Network > Ego-networks > Density path of the UCINET6.0 software. |
| Degree Centrality | Measuring the degree to which a tourism node in the network is directly connected to others, inward and outward degree centrality correspond to the node's ability to cluster and diffuse tourism flows, respectively. | $C_{D(in)}(n_i) = \sum_{j=1}^{c} R_{ij(in)}$, $C_{D(out)}(n_i) = \sum_{j=1}^{l} R_{ij(out)}$; $C_{D(in)}(n_i)$ is inward degree centrality, $C_{D(out)}(n_i)$ is inward degree centrality, $R_{ij(in)}$ indicates that there is a directed link from node $j$ to $i$ direction, $R_{ij(out)}$ indicates that there is a directed link from node $i$ to $j$ direction, $l$ is the number of travel festival points in the network. |
| Closeness Centrality | Measuring the ability of a tourism node in the network not to be 'controlled' by other nodes, there are inward and outward approaches to centrality in a directed tourism network. | $C_C(n_i) = \frac{1}{\sum_{j=1}^{l} d(n_i,n_j)}$; $C_C(n_i)$ is the closeness centrality, $d(n_i,n_j)$ is the shortest tour line from $n_i$ to $n_j$. This is the formula for both inward closeness centrality and outward closeness centrality. |
| Betweenness Centrality | Measures the degree of control of tourism flows by tourism nodes in the network. | $C_B(n_i) = \sum_{j}^{l} \sum_{k}^{l} \frac{g_{jk}(n_i)}{g_{jk}}$, $j \neq k \neq i$; $C_B(n_i)$ is the betweenness centrality of tourist node $i$, $g_{jk}(n_i)$ is the number of shortest tour lines from node $j$ to $k$ and through the node $i$ in the network, $g_{jk}$ is the number of shortest tour lines from node $j$ to $k$. |
| Structural Holes | Indicates the ability of a node to control the flow of resources in the tourism network and is generally measured by effective size and constraint. | This is achieved through the Network > Ego networks > structural holes path in UCINET6.0 software. |

### 3.2.2. Weighted Linear Regression

Based on the development trend of the spatial pattern of tourism flow in the Chengdu–Chongqing Economic Circle and a review of a large number of research studies in the literature, four indicators—road density, number of star-rated hotels, number of scenic spots and regional GDP per capita—were selected to form the influencing factor index system of this study under the conditions of data availability and scientific nature.

The current study shows that factors such as road density, number of star-rated hotels, number of scenic spots and regional GDP per capita have a strong linear relationship with the effective size of tourist attraction structure holes. Moreover, the corresponding weights of each factor can respond well to their contribution to the size of attraction structure holes [21,39]. Therefore, the regression model was constructed as follows:

$$y = \alpha + \beta_1 x_1 + \beta_2 x_2 + \beta_3 x_3 + \beta_4 x_4 \tag{1}$$

The regression model developed, where the independent variable $y$ and the dependent variable $x$ are self-defined according to the purpose of the regression, is solved for $\alpha$ and $\beta$ by the values of the independent and dependent variables. Thus, $y$ corresponds to the effective size of the structural hole, $x_1$ to the density of roads, $x_2$ to the number of star hotels, $x_3$ to the number of scenic spots and $x_4$ to the regional GDP per capita value.

### 3.3. Data Processing

#### 3.3.1. Data Sources

Ctrip.com has set up a travel guide section where users publish their travel reports, which can be searched by keywords such as province and city to find the corresponding user travel reports. A custom Python script is used to collect the data, which you can access

the script in the Supplementary Materials. The script can collect the corresponding user travelogues from the Ctrip section according to the keywords entered and the time interval set. For example, if you enter 'Chongqing' and set the time interval from 1 January 2020 to 31 December 2020, you will be able to read and locally save all travelogues published in 2020 that involve Chongqing.

The data were collected by entering the keywords 'Chongqing' and 'Sichuan' in order, and the time interval was set from 1 January 2018 to 31 December 2021. Travel reports published by users on Ctrip.com between 2018 and 2021, which included travel information from Sichuan Province and Chongqing City, were collected. For example, if User A published a travelogue in 2018 and the travelogue included the travel place name of Chongqing City or marked the target travel place as Chongqing, the travelogue was automatically read by the Python program and recognised as a target object for local storage. Using the described method and removing duplicate travelogues, 2607 were obtained.

### 3.3.2. Data Preprocessing

(1) In the first step, the place names appearing in the original travelogues crawled by Python3.9.6 were counted, and all the place names within the scope of Chengdu–Chongqing Economic Circle were screened out using ArcGIS10.5. In the second step, the original travelogues that did not contain tourism information within the Chengdu–Chongqing Economic Circle were eliminated based on the place names filtered in the first step. In the third step, duplicate travelogues were eliminated, and invalid travelogues, such as commercial propaganda, were screened out. In the fourth step, only travelogues that included two or more scenic spots were retained. After the layers of screening, 1773 travelogues were finally obtained, 281 in 2018, 724 in 2019, 394 in 2020 and 374 in 2021.

(2) In this study, the target area was divided into 44 units based on counties or cities. Based on the above steps, firstly, the Python3.9.6 tool was used to convert the travel time, travel title, travel route and other relevant information in all travel texts into data information stored by Excel. The administrative unit to which each scenic spot belonged was determined using Gaode Map API and manual correction. Secondly, a map of the Chengdu–Chongqing Economic Circle was drawn using ArcGIS10.5, and the longitude and latitude corresponding to the centre point of each administrative unit were found and stored in the Excel database using ArcGIS10.5 surface elements to obtain the centre point. Finally, with the help of Python3.9.6, the number of landscape nodes and administrative unit nodes included in the Tourism Digital Footprint information between 2018 and 2021 was counted. The results showed that 182 landscape areas and 35 units were involved in 2018, 255 landscape areas and 37 units in 2019, 260 landscape areas and 33 units in 2020 and 259 landscape areas and 38 units in 2021.

### 3.3.3. Data Processing and Analysis

The first step was to generate a relational network matrix. After some exemplary preprocessing, this section used Python3.9.6 to convert the tourism routes into a directed multivalued matrix format that can be read by UCINET6.0, according to the flow between scenic areas and administrative units. On this basis, the matrices were imported into the UCINET6.0 software for further analysis.

In the second step, the tourism flow network structure indicators were calculated. This study's tourism flow network structure indicators include overall network density, the centrality of tourism nodes and structural holes. In calculating the overall network density, this section was completed by the path Network > Ego-networks > Density of the UCINET6.0 software. When evaluating the indicators of tourism nodes, this study started the analysis from the centrality of tourism nodes and structural holes, and this part was analysed only with scenic tourism spots as nodes, in which the analysis of the centrality of tourism nodes

involves three specific indicators: degree, closeness and betweenness centrality. The structural holes of the tourism nodes were analysed in terms of the effective size of the nodes.

In the third step, a spatial visualisation analysis was carried out to study more intuitively the characteristics of the evolution of the spatial pattern of tourism flows in the Chengdu–Chongqing Economic Circle. Firstly, the Gephi0.9.5 tool was used to visualise the network representation of digital footprints with tourist attractions as nodes in the Chengdu–Chongqing Economic Circle from 2018 to 2021. For the aesthetics of data presentation, only nodes with a degree sum of 30 and above are shown in the figure. Second, ArcGIS10.5 was used to visualise the digital footprints of administrative units, and geographic data information was added to show the distribution pattern of tourism flows in the Chengdu–Chongqing Economic Circle in terms of geographic space.

## 4. Study on the Evolving Characteristics of the Spatial Pattern of Tourism Flows in the Chengdu–Chongqing Economic Circle

### 4.1. Evolution of the Overall Network Structure of Tourism Flows

Tourism flow is the sum of the number of tourists entering and leaving a place within a certain period. The Gephi0.9.5 tool was used to construct the tourism flow network structure of the Chengdu–Chongqing Economic Circle from 2018 to 2021, as shown in Figure 2. In the same graph, the circular area indicates the relative size of the sum of inflows and outflows, and the thickness of the lines with arrows between nodes indicates the relative size of directed inflows, with thicker connecting lines indicating stronger connections between nodes.

Figure 2 shows that tourism flows in the Chengdu–Chongqing Economic Circle showed a strong 'dual core' polarisation effect, with more popular scenic spots concentrated in the urban centres of Chengdu and Chongqing over the four-year period. To some extent, this indicates that the administrative centres of the administrative regions have a high concentration effect on tourism flows, probably because the administrative centres are the centres of concentration of resources in a province or city, with a higher level of urban construction. In 2018, Chengdu had more high-volume scenic spots than Chongqing, and the density of links between scenic spots in Chengdu was stronger. However, in 2019–2021, the number of high-volume scenic spots and the density of links between high-volume scenic spots in Chongqing exceeded that of Chengdu. The distribution of tourism flows within the Chengdu–Chongqing Economic Circle is heavily influenced by administrative divisions, with closer links within the same administrative unit and sparser links between administrative units. The density of tourism flow networks within the Chengdu–Chongqing Economic Circle increased significantly between 2018 and 2019. Compared with 2019, the density of the tourism flow network decreased significantly between 2020 and 2021.

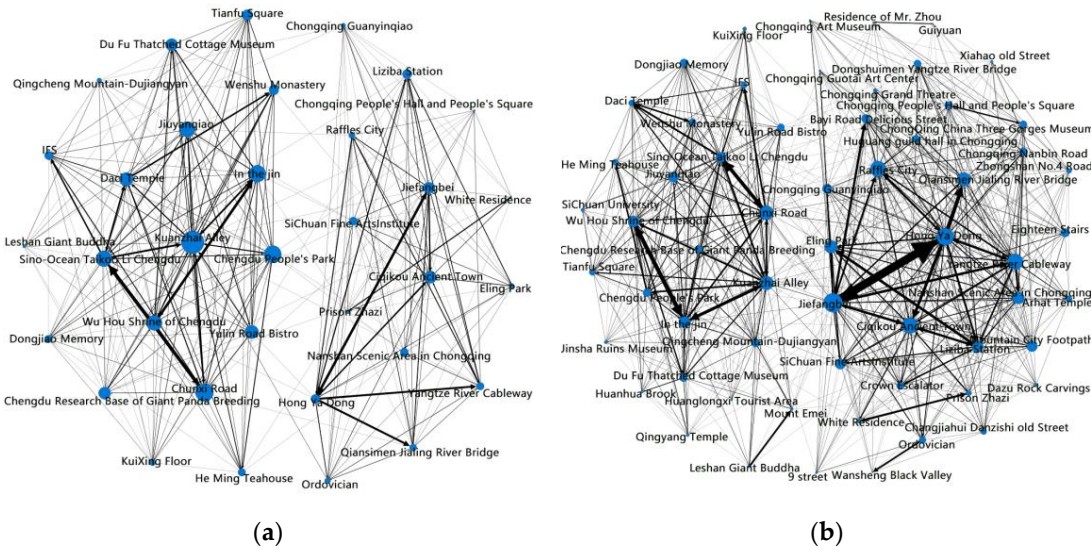

(**a**)                          (**b**)

**Figure 2.** *Cont.*

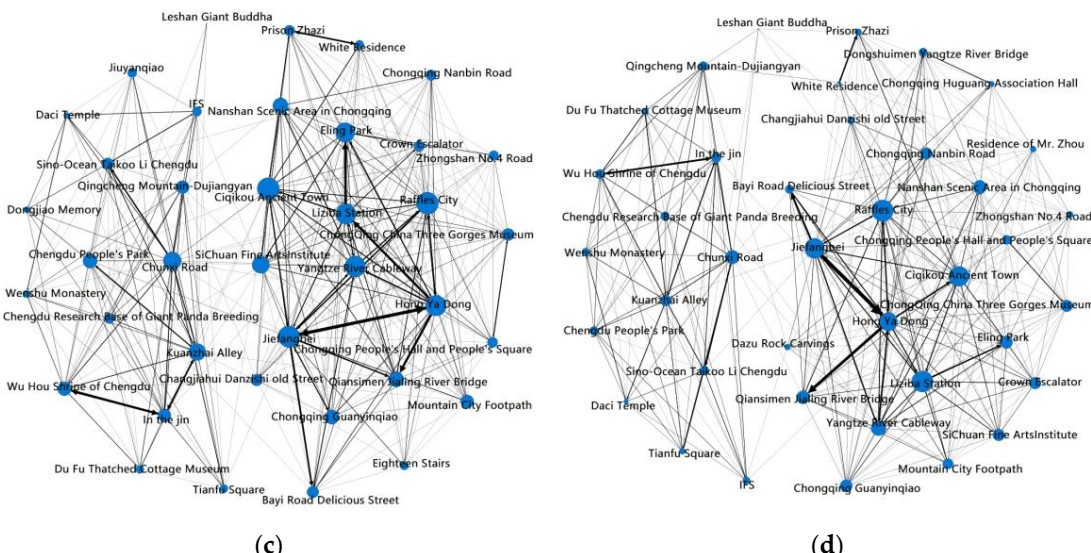

**Figure 2.** (**a**) Tourism flow network diagram 2018; (**b**) tourism flow network diagram 2019; (**c**) tourism flow network diagram 2020; (**d**) tourism flow network diagram 2021.

The overall network density of tourism flows in the Chengdu–Chongqing Economic Circle from 2018 to 2021 is shown in Table 2, with 2019 as the turning point. The distribution of the overall network density values is an irregular inverted 'U' shape (as shown in Figure 3), with overall low network density values for tourism flows within the Chengdu–Chongqing Economic Circle, and an extremely uneven distribution of tourism flows within the circle.

**Table 2.** Overall network density of tourism flows.

|  | **2018** | **2019** | **2020** | **2021** |
|---|---|---|---|---|
| Tourist Attractions—Overall Net Density | 0.0588 | 0.0720 | 0.0342 | 0.0304 |

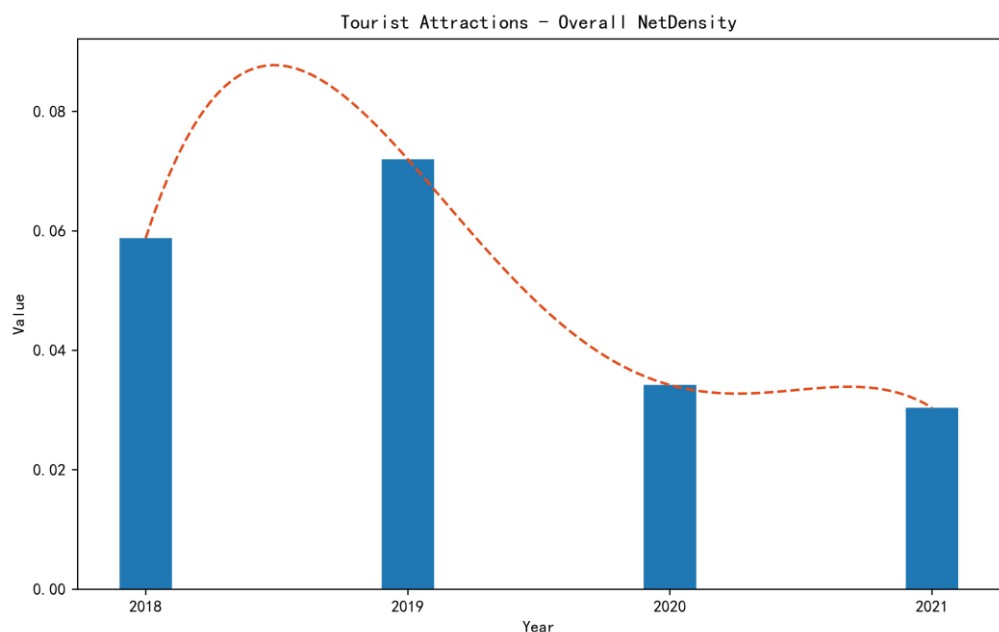

**Figure 3.** Irregular inverted 'U' trend in the overall network density variation of tourism flows.

ArcGIS10.5 was used to map the geographical spatial tourism flow network with administrative units as nodes, as shown in Figure 4. The area of the circle indicates the

relative size of the sum of inflow and outflow, and the thickness of the line with arrows between the nodes indicates the relative size of the directed inflow. The thicker the line, the closer the connection between the nodes.

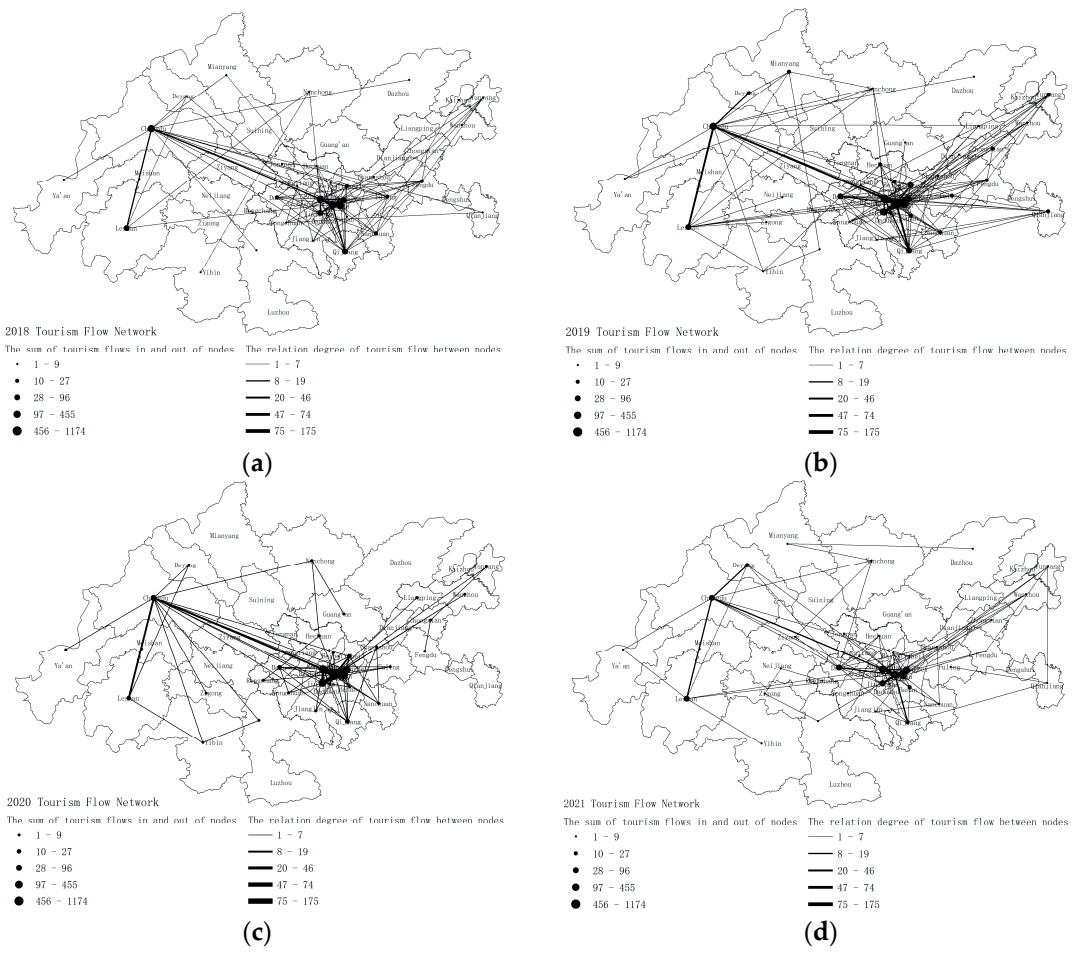

**Figure 4.** (**a**) Network of regional spatial tourism flows in 2018; (**b**) network of regional spatial tourism flows in 2019; (**c**) network of regional spatial tourism flows in 2020; (**d**) network of regional spatial tourism flows in 2021.

As seen in Figure 4, from 2018 to 2019, the geographical scale of tourism flow distribution in the Chengdu–Chongqing Economic Circle was expanding, and the connections between nodes were tightening. From 2019 to 2020, the geographical scale of the overall network was shrinking, and the tightness of the connections between nodes was decreasing. From 2020 to 2021, the geographical scale of the overall network was expanding, and the connections between nodes were slightly decreasing. In the spatial tourism flow network map, Chengdu and Chongqing's central urban areas are the 'twin cores'. Around Chengdu and Chongqing's central urban areas, Leshan City, Nanchong City, northeast Chongqing and southeast Chongqing urban agglomerations form network nodes in four directions.

### 4.2. Node Structure Evolution of Tourism Flow Network

The structural evolution characteristics of tourism flow network nodes were explored by calculating the centrality and structural holes of tourism nodes, where degree centrality, closeness centrality and betweenness centrality were included in the study of tourism node centrality. Structural holes are expressed in terms of the effective size and constraint of nodes.

### 4.2.1. Degree Centrality

Degree centrality is a measure of the degree to which a node in the network is directly related to others, and its magnitude can reflect the strength of the tourism node's ability to collect and disperse in the overall network, where inward degree centrality corresponds to the ability of a tourism node in the network to collect tourism flows and outward degree centrality corresponds to the ability of a tourism node in the network to disperse tourism flows. The results of the degree centrality calculation for each node in the Chengdu–Chongqing Economic Circle tourism flow network are shown in Figure 5. For the sake of aesthetics and readability of the figure, Figure 5 only shows the top 15 nodes in terms of degree centrality in the four years.

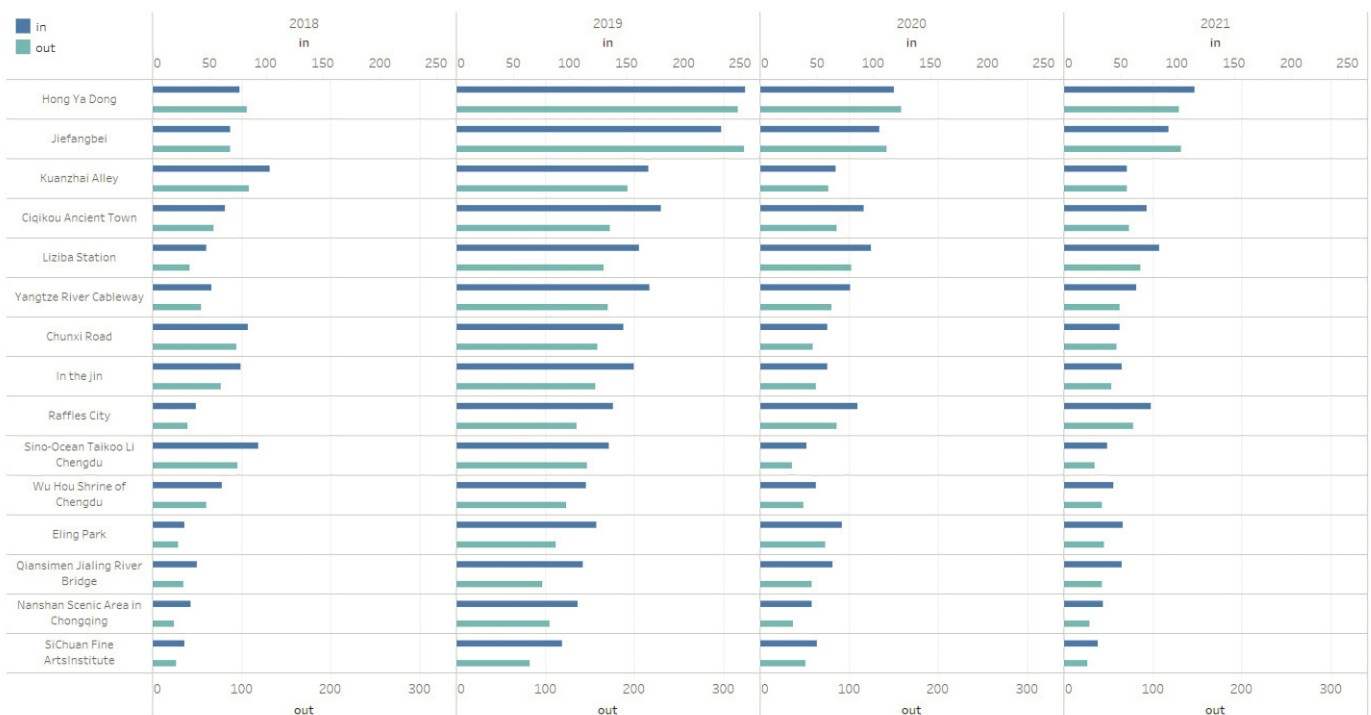

**Figure 5.** Degree centrality value of tourist attractions in the Chengdu–Chongqing Economic Circle in 2018–2021.

By calculation, the scenic spots with the sum of inward and outward degree centrality ranking in the top 20 in all four years are Kuanzhai Alley, Sino-Ocean Taikoo Li Chengdu, Hong Ya Dong, Chunxi Road, Jiefangbei, In the Jin, Ciqikou Ancient Town, Wu Hou Shrine of Chengdu, Yangtze River Cableway, Liziba Station and Raffles City, which are the distribution centres within the Chengdu–Chongqing Economic Circle. Comparing the inward and outward degree centrality of nodes, the diffusion effect of nodes was mostly higher than the agglomeration effect in such centralised scenic spots as Jiefangbei and Hong Ya Dong, which are mostly located in urban centres or transport hubs. Attractive scenic spots with inward centrality higher than outward centrality included Jiu-eye Bridge and Nanshan Scenic Area in Chongqing. Balanced scenic spots with inward and outward centrality close to each other included In the Jin and Wu Hou Shrine in Chengdu, and scenic diversion spots with outward direction greater than inward direction included Jiefangbei and Kuanzhai Alley. Some scenic spots had different types of gathering and dispersal in different years, such as Emeishan in 2018–2019 as an attractive scenic spot and in 2020–2021 as a diversion.

Meanwhile, the mean and standard deviation of degree centrality were much higher in 2019 than in other years, indicating a greater concentration of tourist movement between core scenic areas in 2019. The mean and standard deviation of degree centrality increased

rapidly in 2018–2019, indicating an increase in the average connectivity between scenic areas and a sharp increase in the clustering of tourist trips during this period. In 2019–2020, the mean and standard deviation of degree centrality decreased sharply, indicating a decrease in the average degree of connectivity between scenic areas and a dispersal of tourist trips. In 2020–2021, the mean and standard deviation of degree centrality decreased slightly, the average degree of connection between scenic spots decreased, and tourist trips developed from scattered to slightly clustered.

### 4.2.2. Closeness Centrality

Closeness centrality is a measure of the ability of a tourism node in a network not to be 'controlled' by other nodes. It is divided into inward closeness centrality and outward closeness centrality in a directed tourism network. The results of the closeness centrality calculation for each node in the Chengdu–Chongqing Economic Circle tourism flow network are shown in Figure 6. For the sake of aesthetics and readability of the figure, Figure 6 only shows the top 15 nodes in terms of closeness centrality in the four years.

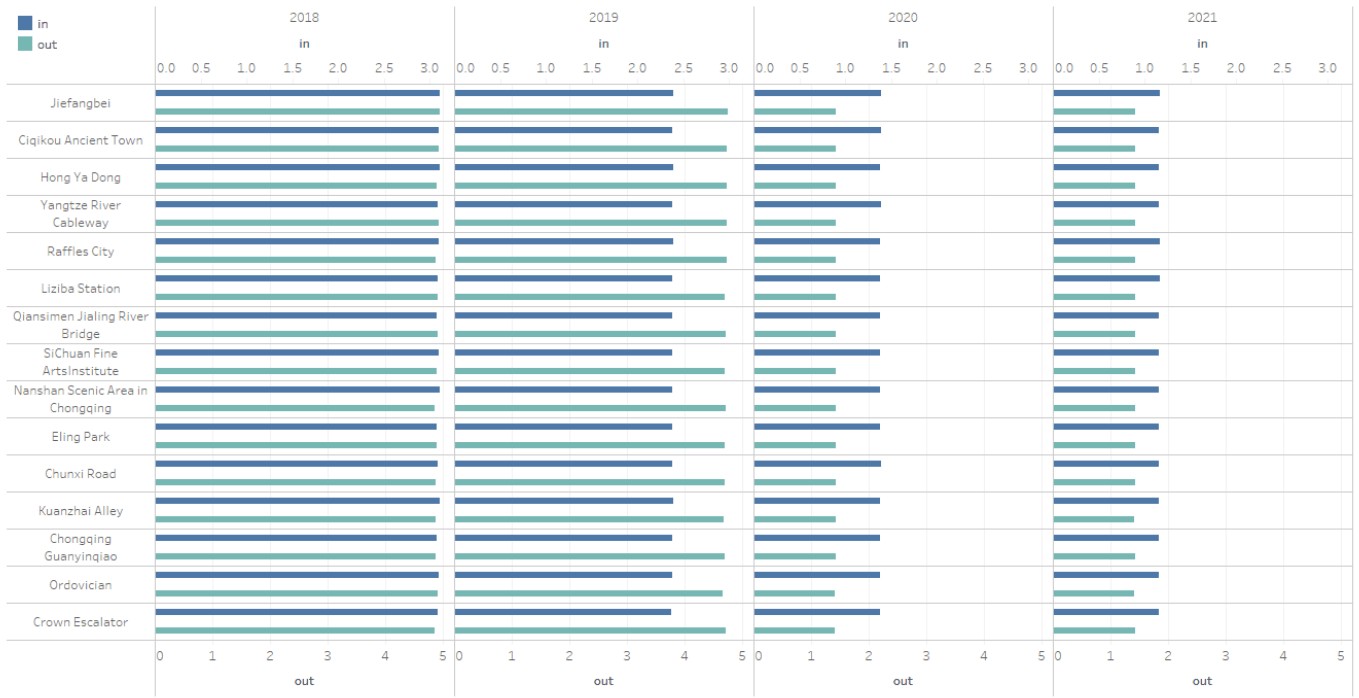

**Figure 6.** Closeness centrality value of tourist attractions in the Chengdu–Chongqing Economic Circle in 2018–2021.

By calculation, the scenic spots with the sum of inward and outward closeness centrality ranking in the top 20 in all four years were Jiefangbei, Ciqikou ancient town, Yangtze River Cableway, Liziba Station, Hong Ya Dong, Sichuan Fine Arts Institute, Kuanzhai Alley, Qiansimen Jialing River Bridge, Raffles City, Eling Park, Chunxi Road, Nanshan Scenic Area in Chongqing and Chongqing Guanyinqiao, which have strong independence and are popular scenic spots on the tourist routes within the economic circle of Chengdu and Chongqing.

In addition, most top-ranked nodes had higher outward closeness centrality than inward closeness centrality, indicating that the most popular scenic areas had higher connectivity convenience as a departure point than an entry point. The mean and standard deviation of scenic area closeness centrality decreased in 2018–2021, indicating an increasing trend of travellers moving from single to multiple scenic areas.

### 4.2.3. Betweenness Centrality

Betweenness centrality corresponds to the degree of control of tourism flows by tourism nodes in the tourism flow network. The results of the betweenness centrality calculation for each node in the Chengdu–Chongqing Economic Circle tourism flow network are shown in Figure 7. For the sake of aesthetics and readability of the figure, Figure 7 only shows the top 15 nodes in terms of betweenness centrality in the four years.

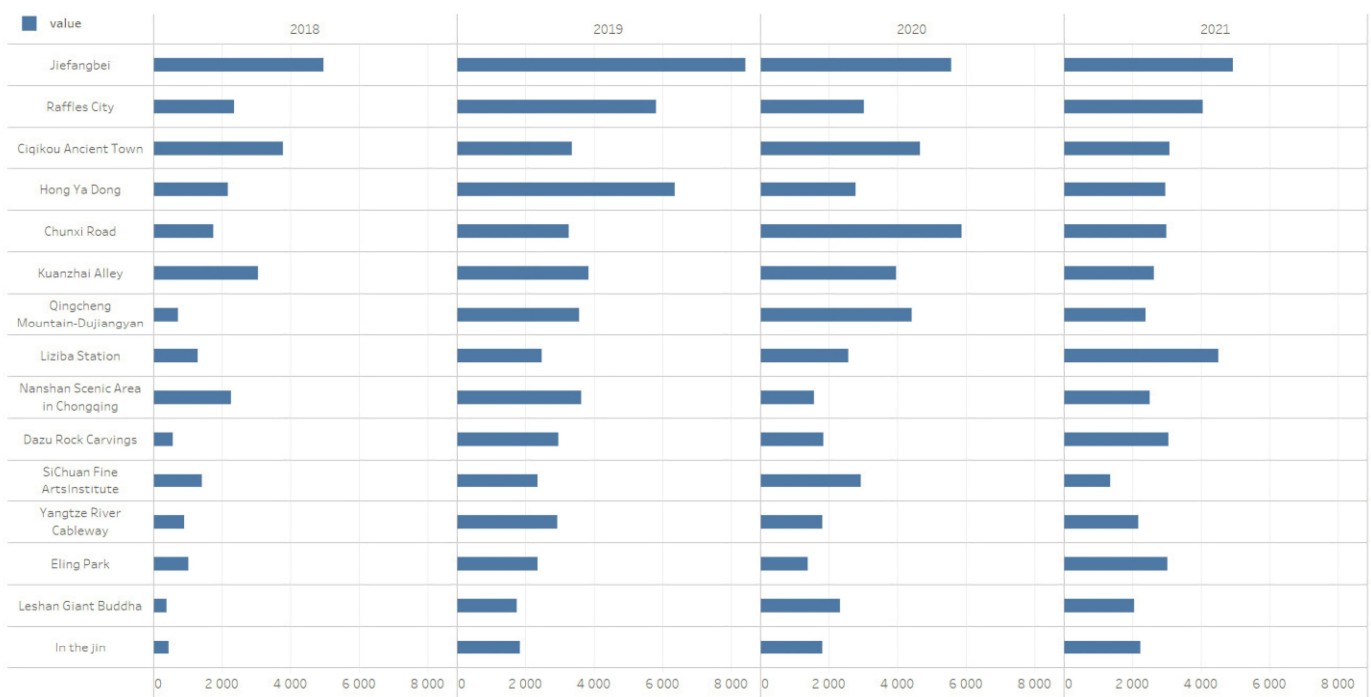

**Figure 7.** Betweenness centrality value of tourist attractions in the Chengdu–Chongqing Economic Circle in 2018–2021.

By calculation, betweenness centrality from 2018 to 2021 was ranked in the top 20 scenic areas, which were Jiefangbei, Ciqikou Ancient Town, Kuanzhai Alley, Raffles City, Nanshan Scenic Area in Chongqing, Hong Ya Dong, Chunxi Road, Sichuan Fine Arts Institute, Liziba Station and Yangtze River Cableway, which have a high degree of control over the tourism flow network, with most tourism flows using these scenic spots as a transit to other scenic areas.

The betweenness centrality of the nodes was highly variable from 2018 to 2021. The scenic spots at the top of the ranking were mainly located in the central urban areas of Chengdu and Chongqing, which are mostly surrounded by convenient transport, are located in commercial centres or have strong scenic features. Scenic spots, such as Qingcheng Mountain-Dujiangyan and Dazu Rock Carvings, which have world-class tourism resources, were ranked relatively highly, although their centrality fluctuates yearly. Many scenic spots, such as Chongqing Baiheliang Underwater Museum Scenic Spot and Leshan Heizhugou Scenic Spot, are usually connected to only a few neighbouring tourism nodes owing to their lack of obvious resource characteristics and marginal position in the tourism network.

### 4.2.4. Structural Holes

The larger the effective size of a structural hole, the stronger its ability to bridge other scenic spots in the tourism flow network that are not directly connected. The results of the effective size and constraint of the structural hole calculation for each node in the Chengdu–Chongqing Economic Circle tourism flow network are shown in Figure 8. For the sake of aesthetics and readability of the figure, Figure 8 only shows the top 15 nodes in terms of the effective size of the structural hole in the four years.

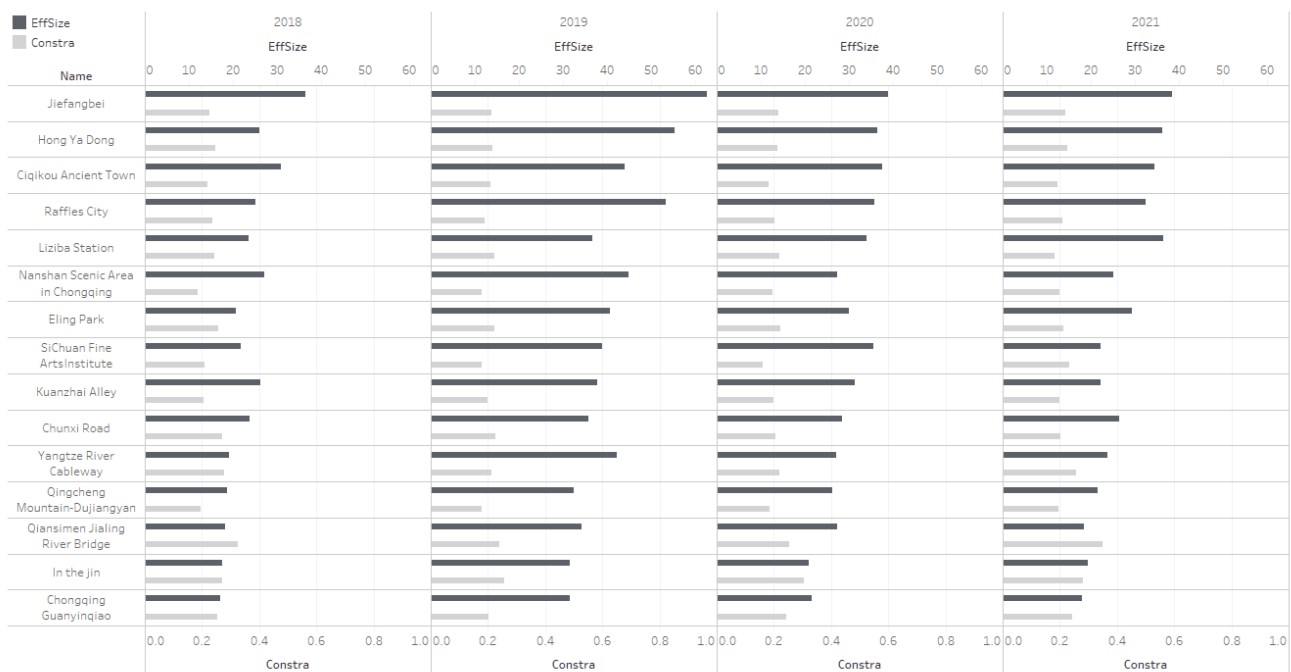

**Figure 8.** Structural holes of tourist attractions in the Chengdu–Chongqing Economic Circle in 2018–2021.

By calculation, in 2018–2021, the scenic spots with the effective size of their structural hole ranking in the top 20 were Jiefangbei, Ciqikou Ancient Town, Nanshan Scenic Area in Chongqing, Kuanzhai Alley, Hong Ya Dong, Raffles City, Chunxi Road, Liziba Station, Sichuan Fine Arts Institute, Eling Park, Yangtze River Cableway, Qingcheng Mountain-Dujiangyan and Qiansimen Jialing River Bridge. These scenic spots had a structural hole much higher than the average, and the constraint was much lower than the average, which created an irreplaceable competitive advantage in the tourism flow network and can link the overall network of tourism flow. However, such tourist attractions are also prone to the over-agglomeration phenomenon.

In addition, the effective size of structural holes in some scenic spots was higher than average, and the constraint was slightly lower. These scenic spots, such as Jiuyanqiao, are important in the Chengdu–Chongqing Economic Circle, which can focus on planning future development to allow them to become core scenic spots while relieving the pressure on core scenic spots. In some scenic spots, the effective size and the constraint of structural holes were smaller than average, such as that of the Chongqing Yuzhong Jiaxi Village Yiyuan scenic spot. These scenic spots are less attractive and less well-known and only play the role of nodes to enrich the tourism flow network of the Chengdu–Chongqing Economic Circle. Some scenic spots had structural holes whose effective size was smaller than average and constraint was larger than the average value, such as the Chengdu Luhu Water City scenic spot. These scenic spots are at the edge area of the overall tourism flow network and have no advantage of structural holes.

## 5. Analysis of Influencing Factors of Spatial Pattern Development of Tourism Flow in Chengdu–Chongqing Economic Circle

### 5.1. Index Selection

Based on the research results of previous scholars [20–22] combined with the development characteristics of the spatial pattern of tourism flow analysed above and the scientific nature and accessibility of data, a regression analysis of the influencing factors of four dimensions (i.e., transport accessibility, tourism reception facilities, tourism resource endowment and economic development level) was carried out. The motorway density, the number of star-rated hotels, the number of high-quality scenic spots rated 3A and above,

and the regional GDP per capita were taken as the corresponding research indicators of the four dimensions. The structural hole efficiency of each administrative unit of the urban agglomeration was taken as the dependent variable.

### 5.2. Source and Processing of Original Data

The research data were obtained from the Chongqing Bureau of Statistics (http://tjj.cq.gov.cn/ (accessed on 1 February 2022)), the Sichuan Provincial Bureau of Statistics (http://tjj.sc.gov.cn/ (accessed on 1 February 2022)), the Chongqing Municipal Commission of Culture and Tourism Development (http://whlyw.cq.gov.cn/ (accessed on 1 February 2022)), the Sichuan Provincial Department of Culture and Tourism (http://wlt.sc.gov.cn/ (accessed on 1 February 2022)) and the government websites or tourism bureau websites of each district, county, city or province. The efficiency value of each administrative unit's structural hole was the effective size of the administrative unit's structural hole. Road density was calculated by the known road mileage of the administrative unit and the area of the administrative unit. The reason for this was that the Yuzhong District road is divided into municipal roads, with no road mileage data. Thus, the regression model did not include Yuzhong data. As of May 2022, the official channel had not updated each administrative unit's road mileage and regional GDP per capita data in 2021. Thus, this regression model only analysed the data for 2018, 2019 and 2020.

### 5.3. Regression Results

Figure 9 shows the fitting effect of the linear regression model. As shown in the legend, the solid blue line indicates the actual effective size of the structure hole, and the dashed orange line indicates the predicted effective size of the structure hole by the regression model. The detailed parameters of the regression model are shown in Table 3. $x_1$ corresponds to road density, $x_2$ to the number of star-rated hotels, $x_3$ to the number of scenic spots and $x_4$ to the value of regional GDP per capita. The corrected $R_2$ value is 0.514; combined with the trend line in Figure 9, the model fits well. $t$ is used to measure the significance of the coefficient statistics, and $P > |t|$ less than 0.05 indicates a significant association between the independent and dependent variables. The indicators of the degree [0.025, 0.975] are the lower and upper limits of the 95% confidence interval.

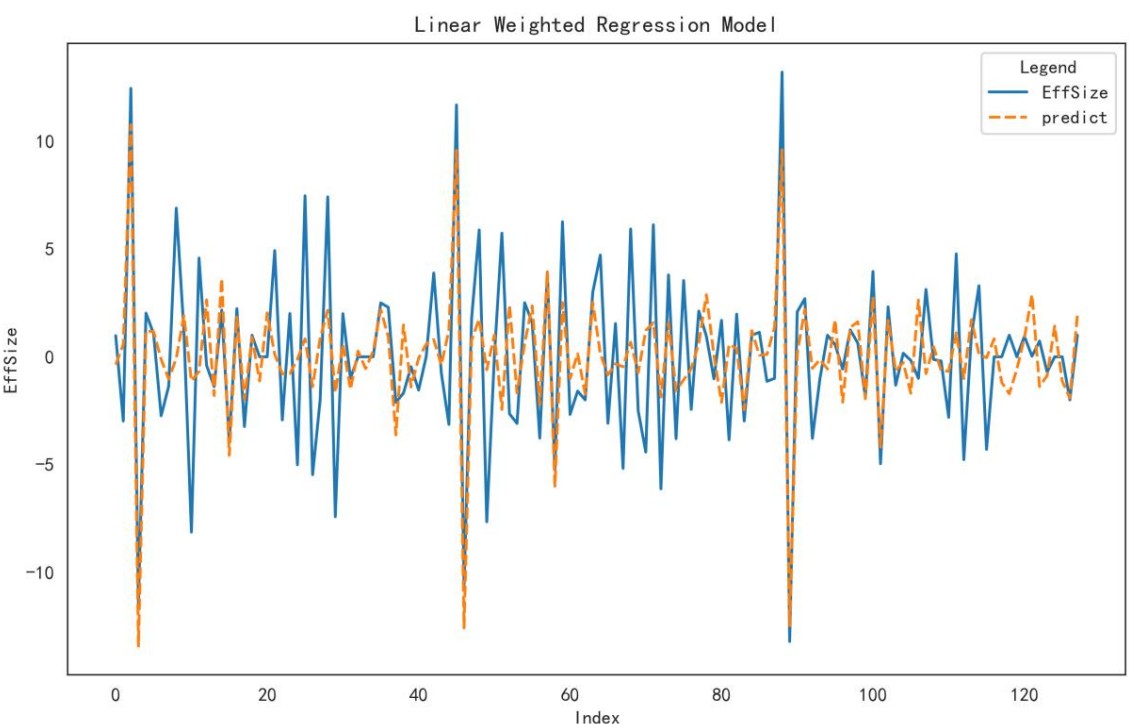

**Figure 9.** Linear weighted regression model.

**Table 3.** Regression parameters.

| | Coefficient | Standard Deviation | $t$ | $P > \lvert t \rvert$ | [0.025 | 0.975] |
|---|---|---|---|---|---|---|
| $x_1$ | 0.2374 | 0.069 | 3.425 | 0.001 | 0.100 | 0.375 |
| $x_2$ | 0.6779 | 0.118 | 5.749 | 0.000 | 0.444 | 0.911 |
| $x_3$ | $-0.0643$ | 0.120 | $-0.535$ | 0.593 | $-0.302$ | 0.174 |
| $x_4$ | 0.3031 | 0.067 | 4.521 | 0.000 | 0.170 | 0.436 |
| Constant term | $1.041 \times 10^{-17}$ | 0.062 | $1.68 \times 10^{-16}$ | 1.000 | $-0.122$ | 0.122 |

The analysis of the data in Table 3 shows that the road density, the number of star hotels and the regional GDP per capita were positively and significantly related to the effective size of the structural hole of the administrative units. The number of high-quality scenic spots was not significantly related to the effective size of the structural hole of the administrative units. Moreover, the constant value of this adjustment result is extremely small and negligible.

The above analysis shows that the better the transport accessibility, the better the tourism reception facilities, and the higher the economic development level of the administrative unit in the Chengdu–Chongqing Economic Circle, the larger the effective size of the structural hole of the administrative unit. Furthermore, the degree of perfect tourism node facilities had the greatest influence on the structural hole effectiveness of the region, followed by regional GDP per capita and transport access. Moreover, no significant correlation was found between the number of high-quality scenic spots and the structural hole effectiveness of administrative units. Therefore, the region's structural effectiveness is not great when it has more high-quality tourist attractions. The laws explain why the tourism flow network in the Chengdu–Chongqing Economic Circle forms a 'dual core' distribution structure.

## 6. Research Findings and Discussion

### 6.1. Research Findings

The study used the Python web crawler function to collect data on the 'tourism digital footprint' of the Chengdu–Chongqing Economic Circle from 2018 to 2021 from Ctrip.com. It also used UCINET6.0 and ArcGIS10.5 to conduct social network analysis and geospatial visualisation of a 'tourism digital footprint'. On this basis, the study used the linear regression weighting method combined with qualitative analysis to study the influence of four indicators, namely, road density, number of star hotels, number of boutique scenic spots and gross regional product, on the effective size of the administrative unit structure hole. Finally, regarding the current development of the tourism industry in the city cluster, the article puts forward four suggestions for the future development of the tourism industry in the Chengdu and Chongqing Economic Circle. The main findings of this study are as follows:

(1) The overall tourism flow network of the Chengdu–Chongqing Economic Circle has a 'dual core' structure consisting of Chengdu and Chongqing central city, with Chengdu occupying the main core position in 2018 and Chongqing central city occupying the main core position from 2019 to 2021.

(2) From 2018 to 2021, the overall net density value of tourism flows in the Chengdu–Chongqing Economic Circle was low, and the density distribution was extremely uneven. With the end of 2019 as the turning point, tourism flow's overall net density value increased and then decreased, with irregular inverted 'U' distribution characteristics.

(3) The tourism flow of the Chengdu–Chongqing Economic Circle has 'two core, four fulcrum' development advantages. The 'two cores' are Chengdu city and the central city of Chongqing, and the 'four fulcrums' are Leshan city, Nanchong city, northeast Chongqing and southeast Chongqing.

(4) Kuanzhai Alley, Hong Ya Dong, Chunxi Road, Jiefangbei, Ciqikou Ancient Town, Yangtze River Cableway, Liziba Station and Raffles City ranked in the top 20 in

centrality and structural hole measurements in the four years and had irreplaceable competitive advantage in the tourism flow network.

(5) In the administrative unit formed by the city, state, district and county, the node structural hole effectiveness is positively influenced by regional transport accessibility, perfect tourism reception facilities and economic development level. A large number of regional high-quality scenic spots do not bring significant benefits to the tourism industry development of the administrative unit.

*6.2. Strategies to Promote Tourism Development in the Chengdu–Chongqing Economic Circle*

The research reveals the distribution and flow pattern of tourism flow in the Chengdu–Chongqing Economic Circle and its development influencing factors, and the research results provide reference significance for the future construction and planning layout of the Chengdu–Chongqing Economic Circle, which should give full play to the advantages of the 'twin core' structure in development and promote the synergistic development of tourism industry of urban clusters in inland areas of western China. Moreover, the cities and counties within the economic circle should fully consider their potential advantages in the network structure in their development and take the initiative to promote regional construction and development. Based on the study's findings, the following four strategies are proposed for the future development of tourism in the Chengdu–Chongqing Economic Circle.

6.2.1. Making Full Use of the 'Dual Core' Advantage to Promote the Simultaneous Development of Several Regions

According to the findings in Articles 1, 3, 4 and 5, the future development of the tourism industry in the Chengdu–Chongqing Economic Circle should take advantage of the 'dual core' effect formed by the central urban areas of Chengdu and Chongqing and reasonably plan to take Chengdu and Chongqing central urban areas as two cores, with Leshan City, Nanchong City, Northeast Chongqing and Southeast Chongqing as four fulcrums. It should also promote the multiregional synergetic development of the tourism industry in the one-pointed city group and take the 'one-hour circle' of the Chengdu–Chongqing city group as an opportunity to accelerate tourism links between administrative units.

For example, based on the further strengthening of the connection between the scenic spots in Chengdu and the central city of Chongqing, the core scenic spots such as Jiefangbei, Hong Ya Dong and Kuanzhai Alley should give full play to the radiation of the peripheral scenic spots and build a cooperative tourism belt for the Chengdu–Chongqing urban agglomeration through joint marketing and other means. The 'pivot point' effect of Leshan city, Nanchong city, northeast Chongqing and southeast Chongqing urban agglomeration should be strengthened and enhanced, and the traffic construction between the 'pivot point' and the 'dual core' should be improved to enhance the convenience of tourism flow. As for the 'four pivot points', while taking over the tourism flow of the core tourism area, attention should be paid to constructing the road network to improve the tourism connection with the neighbouring administrative units. The opportunity should be seized to build the 'one-hour circle' of the Chengdu–Chongqing city group. Chengdu is the centre and includes Leshan City, Mianyang City, Meishan City, Sui City and Chongqing Municipality. The city of Chengdu is the centre and includes the cities of Leshan, Mianyang, Meishan, Suining and Neijiang traffic circles. The central city of Chongqing is the centre, including the group of Bishan District, Hechuan District, Yongchuan District, Dazu District, Changshou District, Tongnan District, Rongchang District, Fuling District, Fengdu County and Dianjiang District traffic circle. The central city relies on the 'one-hour traffic circle' for traffic tourism planning, and the orderly development of road tourism. Other nodes should also fully exploit their advantages in the tourism network and establish and strengthen their links with other nodes in the tourism flow network by improving tourism reception capacity and traffic accessibility.

6.2.2. Reasonable Planning of the Integrated Development Path of Tourism in the Chengdu–Chongqing Economic Circle

Based on the results of study no. 2, the city cluster should fully grasp the policy advantages and explore reasonable ways to reduce the restrictions caused by administrative barriers to the integrated and synergistic development of the region, especially by making good use of elements such as transportation planning within the city cluster, the layout of tourism facilities and regional economic construction to promote the integrated development of the tourism industry steadily.

For example, the Chengdu–Chongqing Economic Circle could be taken as a unit to build an integrated tourism development guidance platform and establish a tourism development committee to solve common problems in developing the tourism industry in the city cluster in a timely manner. It can rely on internet technology to build a tourism industry integration monitoring platform and connect with the tourism industry data statistics centre of each city and county, improve the provision of technical support, and build an integrated tourism service website of the Chengdu–Chongqing Economic Circle, marketing scenic tourist spots in the Chengdu–Chongqing Economic Circle on a large scale. The circle can formulate a standardised service level and integration realisation path, make a top-level design for the integrated construction of the tourism industry in the Chengdu–Chongqing Economic Circle and analyse the characteristics and advantages and role positioning of each administrative unit in the economic circle in combination with the characteristics of spatial pattern development of tourism flow. It can implement the major urban cluster transport planning, construction of tourism service facilities and layout of commercial centres and promote the establishment of an integrated tourism area within the urban cluster in terms of policy, transport, services and commercial economy.

6.2.3. Orderly Development of the Tourism Industry under the Normalisation of the Epidemic

Based on the results of study no. 2, in 2018–2021, the overall network density value of tourism flow and tourism flow network index values of the Chengdu–Chongqing Economic Circle show an irregular inverted 'U' shape, which may also reflect the negative impact of COVID-19 on the development of the tourism industry to a certain extent. In the future, the Chengdu–Chongqing urban cluster should actively explore how to develop the tourism industry in an orderly manner under the normalisation of the epidemic, adapt to the psychological changes of tourists under the normalisation of the epidemic, promote the development of smart tourism and contactless tourism, promote multiple tourism nodes, advocate the evacuation travel mode and expand the overall network scale of tourism flow in the urban cluster while reducing the mass gathering of tourists.

For example, rational planning and construction of contactless service hotels, contactless reception at scenic spots, online booking and face-brushing to enter can minimise the infection caused by human–human contact and human–object contact in the tourism industry under the normalisation of the epidemic. Moreover, the 'cloud' and live tourism industry should be properly developed, tourism image ambassadors and tourism intellectual property should be created and the development of online tourism should be promoted. The development of the online tourism industry in the Chengdu–Chongqing urban cluster should be accelerated by combining online tourism with agricultural production, handicraft production and non-foreign heritage products in the urban cluster. Steps should be taken to integrate the same type of scenic spots in the Chengdu–Chongqing Economic Circle, promote them in bulk through combining online marketing and offline promotion and to consciously guide tourists to travel nearby and spread out. The focus is on developing tourism within the Chengdu–Chongqing Economic Circle under the normalisation of the epidemic and tapping the surrounding market. Based on the Chengdu–Chongqing Economic Circle, the epidemic prevention policy should be formulated for the cultural and tourism-related sectors of the Chengdu–Chongqing urban cluster, and the epidemic prevention work of the cultural and tourism sectors of the Chengdu–Chongqing urban

cluster standardised by establishing a unified standard to ensure the normalisation of epidemic prevention and promote the development of the tourism industry of the urban cluster in an orderly manner.

6.2.4. Reconstruction of Key Nodes to Help the City Group Maintain Connectivity

Based on the results of study no. 5, the development of tourism in the Chengdu–Chongqing Economic Circle has a certain tendency of grouping, in which the trend of 'dual core' regional grouping consisting of the central city of Chongqing and Chengdu is stronger, but the trend of the cross-regional grouping of other administrative units is weaker. The administrative units in the peripheral areas should integrate their resource advantages and shift from promoting multiple scenic spots to creating highly popular scenic spots or special tourist areas. These units rebuild important hubs through the construction of tourist reception facilities, the layout of commercial centres and regional transport planning to reduce the uneven distribution of tourism flows in the overall network of the Chengdu–Chongqing Economic Circle and help the tourism industry of the city cluster to work together.

For example, Yibin City is rich in tourism resources but lacks core tourist attractions. Thus, it should speed up the establishment of 5A scenic spots, focus on building a core distribution centre, and build a better regional tourism brand and visibility. Taking Yibin City Shu Nan Zhuhai Scenic Area as an example, this scenic area has rich tourism resources and better transportation conditions. If it is made a key recreational scenic area for Yibin City tourism, it can promote the development of interactive experience tourism in the scenic area based on existing resources, focus on exploiting the bamboo cultural characteristics of the scenic area and strengthen the ecological protection of the scenic area. It combines various types of tourism, such as folk festivals, water surfing, bamboo sea adventure, hot springs and recreation with the original bamboo sea to build Yibin into a comprehensive tourism scenic area with ecological tourism as the main focus. Good marketing tools can be used to build Shu Nan Bamboo Sea gradually into a newly promoted core scenic area in the Chengdu–Chongqing Belt and Road Economic Circle to promote the whole of Yibin Municipality and neighbouring administrative units and the tourism industry.

*6.3. Directions for Further Research*

This study explores the evolution of the spatial pattern of tourism flows in the Chengdu–Chongqing Economic Circle and its influencing factors through social network analysis and linear regression methods with the help of a 'tourism digital footprint'. The study proves that social network analysis is a good combination of qualitative and quantitative research, allowing the social system of the Chengdu–Chongqing Economic Circle to be considered as a whole and its constituent parts simultaneously. This study revealed the integrated nature of the Chengdu–Chongqing Economic Circle and discovered the hierarchical nature of the overall network. It can explain the closeness of the links between the various nodes within the economic circle and the relationships between the nodes and provide a more comprehensive study of the Chengdu–Chongqing Economic Circle. However, the social network analysis method has the corresponding limitation that its findings only apply to the group under study and often do not have the significance of statistical inference [36,37]. In addition, as for the data of this study, only the 'tourism digital footprint' of Ctrip.com was collected, and the data cannot fully reflect the 'tourism digital footprint' of the Chengdu–Chongqing Economic Circle. At the same time, given the difficulty and scientific nature of data collection, only four typical indicators were selected in the regression model of this study, and the indicators involved need to be further enriched.

In future research, data can be collected from multiple channels, such as Ctrip.com, Qunar.com and Mafengwo.cn, to improve the comprehensiveness and scientific nature of the data. Analysing the influencing factors can switch from taking the structural hole effectiveness of administrative units as the dependent variable to taking the structural hole

effectiveness of tourist attractions as the dependent variable. Indicators such as the node heat index and marketing efforts of tourist attractions can be added [21] to explore the influencing factors of the evolution of the spatial pattern of tourism flows at a more comprehensive micro scale. In terms of research depth, in the future, we can classify tourists into different types, explore the distribution characteristics of tourism flows formed by different types of tourists, and make suggestions for the development of the tourism industry in the Chengdu–Chongqing Economic Circle in terms of tourists' choice preferences [22].

**Supplementary Materials:** The Python scripts used in this manuscript have been placed on GitHub. See the link for details at https://github.com/Visien/Tourist-Flow-Web-Crawler.

**Author Contributions:** Xuejun Chen is responsible for formulation of research ideas and research design, and revision and improvement of the manuscript; Yang Huang is responsible for literature analysis, manuscript writing, and graph drawing; and Yuesheng Chen is responsible for data analysis and processing. All authors have read and agreed to the published version of the manuscript.

**Funding:** Chongqing Social Science Planning Project under Grant No. 2021YC046, Sichuan Provincial Key Research Base of Philosophy and Social Sciences—Research Center for Sichuan Tourism Development Project under Grant No. LY22-05, Science and Technology Research Program of Chongqing Municipal Education Commission (grant number: KJQN202101625), Technology Foresight and System Innovation Project of Chongqing (grant number: CSTB2022TFII-OIX0081).

**Institutional Review Board Statement:** This research did not involve human or animal subjects.

**Informed Consent Statement:** Not applicable.

**Data Availability Statement:** The research data were obtained from Ctrip.com (https://www.ctrip.com/ (accessed on 15 January 2022)), Chongqing Bureau of Statistics (http://tjj.cq.gov.cn/ (accessed on 1 February 2022)), Sichuan Provincial Bureau of Statistics (http://tjj.sc.gov.cn/ (accessed on 1 February 2022)), Chongqing Municipal Commission of Culture and Tourism Development (http://whlyw.cq.gov.cn/ (accessed on 1 February 2022)), Sichuan Provincial Department of Culture and Tourism (http://wlt.sc.gov.cn/ (accessed on 1 February 2022)), and the government websites or tourism bureau websites of each district, county, city, or state.

**Acknowledgments:** The authors would like to thank all participants in this study. Special thanks to the Chongqing Municipal Bureau of Statistics, the Sichuan Provincial Bureau of Statistics, the Chongqing Municipal Commission of Culture and Tourism Development, the Sichuan Provincial Department of Culture and Tourism, and the governments or tourism bureaus of each district, county, and city and state for providing data support for the study.

**Conflicts of Interest:** The authors declare no conflict of interest.

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
