# Peer review of "Spatial Pattern Evolution and Influencing Factors of Tourism Flow in the Chengdu–Chongqing Economic Circle in China"

_ijgi, doi:10.3390/ijgi12030121_

Round 1
Reviewer 1 Report (Previous Reviewer 2)
All of my comments have been well resolved in this revision.
Author Response
Thank you again for your valuable suggestions on this manuscript and I wish you all the best.

Reviewer 2 Report (Previous Reviewer 3)
The Authors addressed my comments in the revised version of the paper. There are some minor language/formatting problems that could be further improved. Other than that I do not have additional remarks.
Author Response
Thank you again for your valuable comments on this manuscript. This revision has improved the grammar and expression of the manuscript throughout, and I wish you all the best.And I invited a professional institution to revise the language expression and grammar throughout the manuscript.Please see the attachment.

Reviewer 3 Report (New Reviewer)
The paper with title Research on spatial pattern evolution and influencing factors of tourism flow in the Chengdu-Chongqing economic circle in China is interesting. Nevertheless, I have several remarks on the paper. First of all, I had problems following the structure of the paper. It does not correspond with the standard structure. The description of used data is after presenting results; there is no discussion section. The introduction is relatively short, and I am not sure if readers can make a study again if needed. So the description of data and methods should be better and more profound in some aspects. The maps and graphs are small and not detailed—some are in B/W format. I recommend that authors be more precise in describing the methodology and be exact on which data and how it was used. Definitely, there should be a discussion of the weaknesses and advantages of the used methodology and data. The list of literature is also not covering the titles in this domain. Twenty-seven sources are not sufficient for this kind of study. Please be more critical and search for other states and approaches.
Author Response
Thank you for your valuable comments on this manuscript. The following changes have been made to the manuscript as a result of your suggestions.
- The description of data and methods has been added to a large part of the manuscript and the placement of data and results has been rearranged.
- The clarity of the images has been further optimised so that you can see the details of the images clearly when you zoom in on the page.
- A discussion of the entire study section has been added at the end of the manuscript, explaining the strengths and weaknesses of the methods used and looking at future directions for refining the study. I would like to thank you again for your important help in improving this manuscript and wish you all the best.

Reviewer 4 Report (New Reviewer)
The reviewed article deals with interesting issues. However, it has a few shortcomings that should be corrected. I will point them out in the order in which they appear in the text.
1. The abstract is missing a few words about Social network analysis
2. Chapter 3.1 should include a map of the study area. It should include all 27 districts (counties) and 15 cities. The reviewer is unable to follow the correctness of the inference without this map.
3. row. 96 "It is a common method used by scholars to study tourism flows" if this is the case, please provide a minimum of 10 examples is application outside of China.
4. indicators in Table 1 are from the literature. It should be given under the table
5. row. 118 don't understand why "y represents the effective size of structural holes"
6. Section 2.1.2 raises the most concerns for me. Should be significantly improved. especially since the article is to be published in the IJGI. I propose to discuss data preparation in the order in which it is done and the tools used
- please discuss the ctrip application, its structure and where to find "the travel notes including the tourism flow information of Sichuan Province and Chongqing".
- Please provide some examples of statistical objects "The statistical object was the travel notes including the tourism flow information.
- please discuss the web crawler function of the Python. Did the authors develop the python script themselves?
- First of all, there is a lack of discussion on the quality of data (statistical objects). It seems to me that they have both good points and bad aspects. the fact that there were 281 objects for 2018, 724 objects for 2019 and almost 400 for 220 and 2021 alone can project the results and interpretation.
7. Does the diagram - figure 1 - have a geographical reference?
8 . row 161 "The overall 161 network density presents an inverted "U" distribution" It is useful to show this distribution on a histogram.
9. I have doubts that the values of 0.03 in Table 2 are correct. Please check
10. Not all geographical names (row 189 -200) can read on the map (figure 2)
11. Section 6.1. Research findings.
The authors present the results in an arbitrary manner. It's as if they were based on the full statistical community. And this is not the case with this publication. Without reference to the method of social network analysis used. The conclusion will be correct if they take this fact into account.
I wanted to point out that there is probably no obligation to post travel notes in ctrip. For this reason, 2019 cannot be considered a turning point. It is simply that more tourists posted notes and whether there were more is a question.
Author Response
- Thank you for your valuable comments on the manuscript. At your suggestion, a note on social network analysis has been added to the abstract section of the manuscript.
- In Chapter 3.1 of the manuscript, a map of the extent of the study area has been added.
- Ten examples of applications of social network analysis to study tourism flows in countries other than China are presented in the manuscript, as detailed in the references section.
- At the end of Table 1, the references for each specific indicator in the table are given.
- In the regression model built, the independent and dependent variables y and x are self-determined according to the purpose of the study, and the regression model is solved for α and β by the values of the independent and dependent variables. y therefore corresponds to the effective size of the structural hole. a note on this part is also added in this manuscript.
- At your suggestion, this manuscript provides additional clarification on the data processing section, a brief description of the location of the travelogues in Ctrip, examples to illustrate the statistical objects, and a description of the Python script used in this manuscript.
- Regarding your question about the quality of the data, it is clear from the tourism statistics of Sichuan Province and Chongqing that the number of travelogues on Ctrip is consistent with the current state of tourism in the Chengdu-Chongqing twin-city economic circle during this four-year period, i.e. tourism developed best in 2019, so I believe that the data collected in this manuscript is more scientific.
- At the same time, it is almost impossible to present the results and interpretations of this manuscript without conducting social network analysis and regression analysis, as the number of travel logs can indicate the motivation of users to post travel logs each year, i.e. the more travellers there are in a given year, the more likely users are to be involved in posting travel logs. However, the number of travel logs is not necessarily linked to the other research indicators in this manuscript. For example, it is not the case that the higher the number of travel logs in a given year, the higher the value of travel network density in that year; if there are twice as many travel logs in 2019 as in 2018, but the number of route connections has only doubled compared to 2018 and the connection trajectories remain the same, then the travel network density in 2018 is necessarily the same as in 2019.
- The purpose of Figure 2 in the revised manuscript you mention (Figure 1 in the original manuscript) is to show the tourism relationships between the nodes and the presentation characteristics of the overall tourism flow network diagram, so the individual nodes do not correspond to the geographical coordinate positions one by one. However, based on Gephi 0.9.5's automatic identification of groupings and careful verification by the author, it was ensured that the left points of each diagram corresponded to scenic spots in Sichuan Province and the right points to those in Chongqing City.
- At your suggestion, a bar chart showing the changing density of the tourism network has been added to this manuscript.
- After repeated validation, the values in Table 2 are correct.
- I am very sorry that the nine districts of Chongqing city centre are too compact to be visible on the map. However, following your suggestion, I have added a map illustrating the scope of the study - Figure 1 - which has the same base map as Figure 4 and can be viewed with the help of Figure 1.
- Explanations of social network analysis methods and linear regression have been added to the results section of this manuscript.
- Regarding your reference to "2019 cannot be seen as a turning point. It is simply that more tourists posted notes, and whether there were more is a question." The answer to this point is the same as the answer to point 6. The number of travelogues is an indication of how active users are in publishing travelogues in a given year, i.e. the more tourists travel in a given year, the more users are likely to be involved in publishing travelogues. However, the number of travelogues is not linked to the other research indicators in this manuscript. For this description, please refer to the section of this manuscript where the indicators are described. What is important for the results part of this study is the direction of flow of all tourism flows, i.e. the structural characteristics of the network of tourism flows. Therefore, by examining the structure of the network of tourism flows from 2018 to 2021, it is theoretically possible to consider 2019 as a turning point.
Thank you again for your valuable suggestions on this manuscript, from which I have benefited greatly.
Please see the attachment.

Round 2
Reviewer 3 Report (New Reviewer)
I would like to thank the authors for incorporating the remarks into the manuscript. Now the paper is a lot better.
I have one recommendation, will it be possible to add your python scripts(s) as supplementary material or GitHub and include the link to the text?
Author Response
Thanks for your valuable suggestions and help in revising the manuscript。I have made the Python scripts used in this manuscript available on GitHub as supplementary material and put the links before the references.
Best regards.

This manuscript is a resubmission of an earlier submission. The following is a list of the peer review reports and author responses from that submission.
Round 1
Reviewer 1 Report
The paper is well-structured and well-written, however, it overall merit is limited since there is little (or no) evidence of novelty. Although the authors claim that "there are relatively low diachronic studies on national urban agglomerations" (lines 60-61), I have found the following ones:
- Naixia Mou, Yunhao Zheng, Teemu Makkonen, Tengfei Yang, Jinwen(Jimmy) Tang, Yan Song, Tourists’ digital footprint: The spatial patterns of tourist flows in Qingdao, China, Tourism Management, Volume 81, 2020, https://doi.org/10.1016/j.tourman.2020.104151.
- Chang Gan, Mihai Voda, Kai Wang, Lijun Chen, Jun Ye, Spatial network structure of the tourism economy in urban agglomeration: A social network analysis, Journal of Hospitality and Tourism Management, Volume 47, 2021, Pages 124-133, https://doi.org/10.1016/j.jhtm.2021.03.009.
- Yang G, Yang Y, Gong G, Gui Q. The Spatial Network Structure of Tourism Efficiency and Its Influencing Factors in China: A Social Network Analysis. Sustainability. 2022; 14(16): 9921. https://doi.org/10.3390/su14169921
among others.
What all three have in common is: they deal with tourism patterns, in China, with social network theory across time and they are all left out form the references!
Therefore, this paper does not only add in the bibliography in terms of research except that it tackles tourism flows in a specific region of China for four years but also it lacks significant relevant published research in the field.
The findings of the paper are not groundbreaking in the sense that it was expected that tourism flows would be lower for 2020 and 2021, where the world was either in lockdown or dealing with the pandemic on a daily basis. Moreover, the fact that accessibility ease, good tourism infrastructure, and high economic development act as magnets to tourisms is nothing new.
For me, the most significant finding is the "dual-core" structure of the tourism flow network and the authors should attempt to provide explanations for this.
The reference to Table 2 in lines 180-181 (As can be seen from Table 2, the mean and standard deviation of number centrality in 2019 are much higher than those in other years) does not correspond to the actual Table 2 in the document. I believe the authors have forgotten to include the correct table and what is Table 2 in the document should actually become Table 3, which is the result of the regression analysis.
Moreover, the terms "proximity centrality" and "intermediate centrality" are more frequently (and more correctly?) met in the bibliography as "closeness centrality" and "betweenness centrality" respectively. The same holds for "effective scale" and "limit system", which, I believe, are referred by Burt himself (1992) as "effective size" and "constraint" respectively instead. By the way, a reference to Burt is mandatory for the scientific soundness of the paper as it was him who introduced structural holes.
Finally, all notions/terms/concepts belonging to the social network theory and analysis are not defined nor explained to the reader.
For all the above reasons, I would not recommend publication of this paper.
Reviewer 2 Report
This paper discusses a study on the spatial pattern evolution and influencing factors of tourism flow in Chengdu-Chongqing economic circle in China. Even though there is extensive tourism flow-related research, few of them is on Chengdu-Chongqing economic circle. The authors conducted social network analysis on spatial pattern evolution and used linear regression model to analyze the influencing factors.
There are several issues that can be resolved to make the paper better.
1. The problem discussed in the paper is not well-defined. Components of a problem like the input and output are missing.
2. The motivation of the problem is not clear. In the introduction, there is a claim that “exploring the distribution, flow law and flunking factors of tourism flow … has important practical guiding significance for promoting the coordinated development of …”, but the paper does not provide enough evidence to support how the study can help.
3. The reason why social network analysis and linear regression are used is not provided. Are there any other methods that can be used? Can they provide some more interesting results?
4. The literature review does not support the novelty claims of the paper well. The authors listed some work on tourism flow analysis for some other regions as well as some methods to study the influencing factors, but did not present the relationship between these work and this paper. Maybe these work can be elaborated to validate the methods.
5. There is a gap between the findings (section 6.1) and the strategies to promote tourism (section 6.2).
6. The presentation of the paper can be improved. There are many grammar issues in the paper.
Reviewer 3 Report
The paper analyzes factors influencing tourism flow in Chengdu-Chongqing economic circle using tourism flow network analysis and weighted linear regression analysis.
My remarks concerning the paper are the following:
- The Authors discuss very briefly the results of the regression analysis. I feel like a discussion of the obtained regression model would be nice. Maybe adding a diagram depicting the regression results would make the analysis easier and the drawn conclusions more obvious to the reader.
- Conclusions presented in chapter 6 should be more related to the research results presented in chapters 4 and 5. I propose to add references in the conclusions/presented plans showing on which findings they are based.
- Some of the presented figures are unreadable - the font size is too small.